# Contrastive Self-Supervised Learning of Global-Local Audio-Visual Representations

## Abstract

Contrastive self-supervised learning has delivered impressive results in many audio-visual recognition tasks. However, existing approaches optimize for learning either *global* representations useful for high-level understanding tasks such as classification, or *local* representations useful for tasks such as audio-visual source localization and separation. While they produce satisfactory results in their intended downstream scenarios, they often fail to generalize to tasks that they were not originally designed for. In this work, we propose a versatile self-supervised approach to learn audio-visual representations that can generalize to both the tasks which require global semantic information (e.g., classification) and the tasks that require fine-grained spatio-temporal information (e.g. localization). We achieve this by optimizing two cross-modal contrastive objectives that together encourage our model to learn discriminative global-local visual information given audio signals. To show that our approach learns generalizable video representations, we evaluate it on various downstream scenarios including action/sound classification, lip reading, deepfake detection, and sound source localization.

## 1 Introduction

Self-supervised learning aims to learn representations of data that generalize to a large variety of downstream tasks. Recently, contrastive self-supervised learning (CSL) has achieved impressive results on several computer vision tasks (Oord et al., 2018; Hjelm et al., 2018; He et al., 2020; Chen et al., 2020). In CSL, the choice of "views" determines the types of information that the representation captures (Bachman et al., 2019), as the framework learns representations that focus on the shared information between views. It has been demonstrated that the optimal choice of views depends critically on the downstream task (Tian et al., 2020). Therefore, existing works mainly focus on finding different views tailored for the intended downstream tasks. For example, when tailoring views for action classification, Hjelm & Bachman (2020) extends DIM (Hjelm et al., 2018) to the spatio-temporal setting by assuming that global and local information useful for action classification (i.e, global semantics) should be invariant across time and space within a given video. When dealing with multimodal data, several approaches utilize audio-visual correspondence from videos (Morgado et al., 2020). Such a CSL approach is based on an assumption that information needed for audio/video classification should be shared between the two modalities.

Although they achieve impressive results in their intended downstream tasks, existing approaches often fail to generalize to tasks that they were not originally designed for. For example, in lip reading (Chung & Zisserman, 2016), the desired information is the fine-grained spatio-temporal representation around the mouth. However, if we directly apply existing CSL approaches, the shared information across views is that a there is a face, while the useful information, the lip movements, will be suppressed as they are changing across views from the sample clip.

Motivated by this, we propose a versatile CSL approach to learn representations that can generalize to both scenarios that require global representations (e.g., classification) and scenarios that require local representations (e.g., localization) (see Fig. 1). Our approach, which we call *global-local cross-modal* (GLCM) contrastive learning, has four key properties that we assume to be important for our learning objective: 1) observations from the same time span of a video should reflect the same content regardless of modalities; 2) the same observations captured at different time scales can reflect both global and local information; 3) when learning on a local temporal scale, the contrasting

views should only share the time-varying information (e.g. only the moving lip) while ignoring globally invariant information; 4) multi-scale (global-local) observations can be trained jointly in a collaborative way so that representations learned at either scale can be reused.

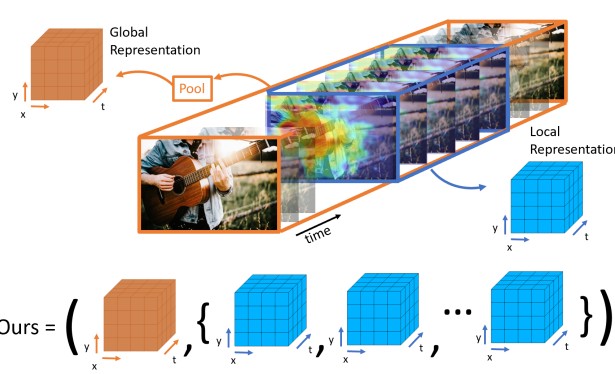

Figure 1: While many self-supervised approaches optimize for high-level *or* low-level tasks, we present an approach to learn both global *and* local representations from video.

We formulate our GLCM objective using two cross-modal contrastive losses computed at multiple temporal scales. Specifically, we generate global and local views of a visual sequence at different sampling rates. The audio sequence is used as an anchor to contrast with global and local visual features, respectively. Consistent with the first property (see above) losses are computed at the same temporal scale, i.e. $z_a^g \leftrightarrow z_v^g$ and $z_a^l \leftrightarrow z_v^l$, such that, given the same video sequence, the learned $z_v^g$ and $z_v^l$ reflect both global and local information; the latter portion satisfies the second property. To implement the third property, the local contrastive loss ($z_a^l \leftrightarrow z_v^l$) is computed by considering only the audio-visual features that lie in the same time window as positive pairs; the others are all negative pairs. Finally, we utilize information captured at the global scale (e.g. localizing the source of a sound) to assist efficient learning at the local scale, thus capturing the fourth property.

We show that GLCM pretraining learns representations with global and fine-grained spatio-temporal information from audio-visual signals. The learned representations perform effectively on a variety of downstream tasks. We evaluate our proposed approach on tasks that needs local spatio-temporal information (i.e lip reading, deep-fake detection, and sound-source localization) and also discriminative tasks that needs global information (i.e. action classification and audio-event classification).

## 2 RELATED WORK

**Contrastive self-supervised learning.** CSL has contributed to strong performance on many tasks and in cases produced comparable results to supervised learning (Chen et al., 2020; Caron et al., 2020). Contrastive learning leverage multiple views of the same data (Hjelm & Bachman, 2020; Oord et al., 2018), e.g., multiple perspectives within the same modality (e.g., augmentations of the same image, different frames of a video, etc.) (He et al., 2020; Hjelm & Bachman, 2020; Han et al., 2019a) or perspectives from different modalities (e.g., depth and RGB images, visual and textual signals) (Tian et al., 2019; Sun et al., 2019; Miech et al., 2020; Alayrac et al., 2020). Chen et al. (2020) and Hjelm et al. (2018) show that leveraging local information to perform contrastive learning further improves the performance on image classification tasks. DIM (Hjelm et al., 2018) has been extended to multi-scale (Bachman et al., 2019) and video data Hjelm & Bachman (2020). However, evaluation is still focused on "discriminative" tasks (image classification and video event classification), while there is little evidence that these models will adapt well to the local information.

**Audio-visual representation learning.** Several approaches have been proposed to leverage the natural correspondence between audio and visual signals to perform CSL (Asano et al., 2020; Korbar et al., 2018; Alwassel et al., 2019; Morgado et al., 2020; Patrick et al., 2020; Chung et al., 2019). Most existing approaches aim to capture high-level semantic information from observations. It has been empirically demonstrated that such learned information is very effective for "discrimination tasks" (classification). However, in tasks that needs local information the learned representations may not perform well. Xiao et al. (2020a) design their approach by utilizing different temporal scales of the audio and visual data, which encourages the model to capture fine-grained temporal information and hence improves the performance. However, the evaluation was limited to classification tasks. In contrast with previous work, we demonstrate that our approach effectively learns global-local audio-visual representations by evaluating on a variety of downstream tasks.

## 3 APPROACH

We propose using the audio and visual channels as cross-modal views of video data. As we aim to learn both local and global temporal information, we utilize the same visual sequence processed at different sampling rates to reflect the same observation at different temporal scales. Given that we want each signal to capture complementary views of the same data, we use different encoders to extract the representations from the audio sequence ($E_a$), subsampled visual sequence ($E_v^g$) and full sampling rate visual sequence ($E_v^l$). The question is, then, how to design a contrastive loss to learn representations from these different views. We achieve this goal by jointly training the model using two contrastive losses: global and local. As shown in Fig. 2, the global loss is computed by contrasting audio signals with subsampled visual sequence (Sec.3.1); while the local loss is computed by contrasting audio signals with visual sequence at a full sampling rate (Sec.3.2). To jointly train the global and local pathways, we propose a spatially-aware attention pooling mechanism to effectively reuse the information that was captured from the global pathway in the local pathway (Sec.3.3).

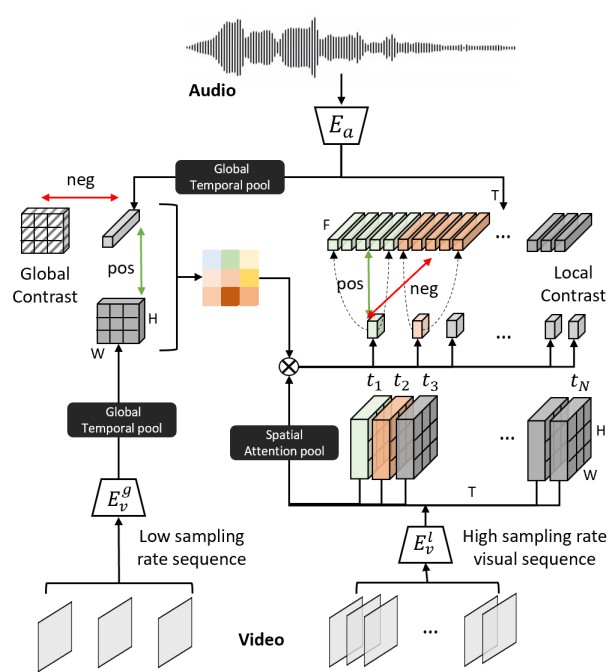

Figure 2: Our GLCM architecture. For clarity we omit the "channel" dimension and only show the spatial and temporal dimensions. In the "global contrast" pathway, the visual features are shown with different shading patterns (e.g., diagonal-hatch, filled gray) and indicate those that come from different video samples. In the "local contrast" pathway, we use colors to indicate different time windows.

### 3.1 GLOBAL CONTRASTIVE OBJECTIVE

We design the global contrastive objective to capture slowly changing information with high audio-visual correlation. We use video sequences captured at low sampling rates, which will inevitably lack local temporal information. $E_a$ encodes an audio sequence into an audio embedding $z_a \in R^{T \times F}$, where $F$ is the frequency, and $T$ is the sequence length. After temporal global pooling, it becomes $z_a \in R^{1 \times F}$. Similarly, we perform global temporal pooling on features encoded by the global visual encoder $E_v^g$, which produces the global visual embedding $z_v^g \in R^{1 \times H \times W \times C}$. Note that, for the visual features, we perform global pooling only along the temporal dimension while keeping the spatial dimension intact. The reason is that when learning in a global temporal space, the model has capacity to capture more spatial information. To compute the global contrastive loss, we consider the audio features $z_a$ and the visual features $z_v^g$ that come from the same video sample as positive pairs, while features coming from different video samples are negative pairs. In order to encourage the model to also capture spatial information, we adopt MIL-NCE (Miech et al., 2020) to compute the loss. Specifically, we consider all $H \times W$ spatial grids in $z_v^g$ as the instances, and therefore, instead of just taking a single audio-visual positive pair $z_a \leftrightarrow z_v^g$, the new positive pair becomes multiple visual instances $z_v^g[i]_{H \times W}$, i.e. $z_a \leftrightarrow z_v^g[i]_{H \times W}$. The loss is then defined as:

$$L_g = -log \left( \frac{\sum_{z_v^g \in \mathcal{P}} exp(z_a^T z_v^g)}{\sum_{z_v^g \in \mathcal{P}} exp(z_a^T z_v^g) + \sum_{z' \in \mathcal{N}} exp(z_a'^T z_v'^g)} \right) \tag{1}$$

where $\mathcal{N}$ is a set of negative audio and visual pairs, $\mathcal{P}$ is a set of spatial grids in $z_v^g$.

## 3.2 Local Contrastive Objective

We design the local contrastive objective to capture fine-grained spatio-temporal information that is sensitive to temporal changes while being invariant to different modalities. We thus contrast between local audio features $z_a$ and local spatio-temporal visual features $z_v^l$. Specifically, we consider the temporal local audio and visual features that lie in the same time window to be the positive pairs, $z_a[t] \leftrightarrow z_v^l[t]$, where $z[t]$ represents features in the time window $t$. As shown in Figure 2, the video and audio features shaded in the same color refer to those in the same time window. The features in different time windows (e.g. green and orange blocks) are considered as negative pairs even if they are from the same video sample. As such, the shared information between the modalities is principally how the features vary over time. We obtain the local audio features by using the same audio encoder $E_a$ but without global temporal pooling. The local visual features $z_v^{local}$ are obtained by feeding the visual sequence with a high sampling rate into the local visual encoder $E_v^l$, which produces the visual features $z_v^l \in R^{T \times H \times W \times C}$. We then perform spatial pooling while keeping the temporal scale the same, the visual features become $z_v^l \in R^{T \times 1 \times 1 \times C}$.

As the audio channel in a video generally has a higher sampling rate than the visual channel, visual feature at a single time step will be mapped to multiple audio feature slices. As shown in Figure 2, at time $t_1$, the visual features $z_v^l[t_1]$ (green block) correspond to multiple audio features $z_a[t_1]_{t \in M}$ (green blocks), where $M = 5$ in Figure 2. Specifically, we use a sliding window of size $M$ to map each set of visual features at a given time step to a window of audio feature slices. Then the positive pair is considered as a visual feature and the corresponding window of audio feature slices. Once again we use MIL-NCE (Miech et al., 2020) to compute the contrastive loss. The reasoning for applying MIL-NCE here is different than in the case of the global contrastive loss. In the global contrastive loss, we aim to let the network capture spatial information. While in the local contrastive objective, the goal of using MIL-NCE is to mitigate the missing strict temporal mapping problem. The loss is therefore defined as:

$$L_l = -log\left(\frac{\sum_{z_a \in \mathcal{Q}} exp(z_a^T z_v^l)}{\sum_{z_a \in \mathcal{Q}} exp(z_a^T z_v^l) + \sum_{z' \in \mathcal{N}} exp(z_a'^T z_v'^l)}\right) \tag{2}$$

where $\mathcal{Q}$ is a set of audio feature slices in the same time window as $z_v^l$, and $\mathcal{N}$ is a set of negative audio and visual pairs.

## 3.3 Spatially-aware attention pooling

As discussed in Sec. 3.1, when computing the global contrastive loss, we focus on the spatial dimension. Therefore, we can utilize spatial information captured from the global pathway to assist the local contrastive loss. Specifically, we compute the correlation (i.e. dot product between the audio feature and each of the visual features in a spatial grid) computed in the global pathway as the attention score; intuitively, this captures the regions of the spatial grid which likely correspond to the source of the sound. For example, in a video of someone talking, the lips will have a relatively higher score when compared to the background, and in a video of someone playing a guitar, fingers on the guitar will have high scores. We thus use the score as an $H \times W$ attention map (see Figure 2). We utilize this attention map to perform spatial attention pooling on local visual features at each time step $P_{atten}(z_v^l[t])$. Comparing with regular spatial average pooling, it helps the network give greater weight to parts within a frame with high audio-visual correspondence. This way, the efficiency of the local contrastive loss can be much improved. We empirically demonstrate the effectiveness of spatial attention pooling mechanism in Table 1 and 2.

## 4 Experiments

**Implementation details.** We use 3D-ResNet18 (Hara et al., 2018) for our visual encoders ($E_v^g$ and $E_v^l$) and 1D-ResNet18 for our audio encoder, in both cases using Batch Normalization (BN) (Ioffe & Szegedy, 2015). All models are trained end-to-end with the ADAM optimizer (Kingma & Ba, 2014) with an initial learning rate $\gamma = 10^{-3}$ after a warm-up period of 500 iterations. We use 16 NVIDIA Tesla P100 GPUs with a batch size of 32 for our experiments. For pretraining, we preprocess video frames by sampling at 10 FPS and applying random cropping, horizontal flipping, gray-scaling, and temporal jittering. We resize video frames to three-channel images of $112 \times 112$; we set the clip

length to 32 frames for the local visual pathway, and a $\frac{1}{8}$ sampling rate for the global path way (8 frames). For the audio channel, we extract mel-spectrograms from the raw waveform using the LibROSA library and get a $80 \times T$ matrix with 80 frequency bands; T is proportional to the length of an audio clip. We then segment the mel-spectrogram according to the corresponding video clips to ensure temporal synchrony. We treat the mel-spectrograms as an 80-channel 1D signal. To compute the global MIL-NCE loss, we use features at a $16 \times 16$ spatial resolution. To compute the local contrastive loss, we adopt a temporal window of size three without overlap.

**Datasets.** Many downstream tasks of interest involve human faces (e.g., deepfake detection), speech (e.g., lip reading) and activity (e.g., audio and video classification). Therefore, we use a combination of Kinetics-700 (Carreira et al., 2019) and AVSpeech (Ephrat et al., 2018) for pretraining. Specifically, we randomly select 120K video samples from each datasets, which gives us a dataset, we term as K-AV, in total 240K samples. For comparison with the other state-of-the-art approaches, we pretrain our model on K-AV dataset which is at the same scale as the Kinetics-700 dataset. For the ablation study, we pretrain our model on a subset of 15K samples from the K-AV dataset, we term as K-AV-15K. As for downstream tasks, we evaluate our models on action recognition using UCF101 (Soomro et al., 2012) and HMDB51 (Kuehne et al., 2011), and on sound classification using ESC50 (Piczak, 2015b). For lip reading, we evaluate our model on both LRW (Chung & Zisserman, 2016) and LRS2 Chung et al. (2017). For deepfake detection, we evaluate our model on a subset of DFDC (Dolhansky et al., 2019).

## 4.1 Downstream Scenarios

| Training | Model | Backbone | Dataset | LRS | LRW |
|---|---|---|---|---|---|
| Superv. | LRW (Chung & Zisserman, 2016) | VGG-M | LRW | N/A | 61.10 |
| | Res. (Stafylakis & Tzimiropoulos, 2017) | ResNet34 | LRW | N/A | 83.00 |
| | TwoStream (Weng & Kitani, 2019) | I3D | LRW | N/A | 84.07 |
| | DFTN (Xiao et al., 2020b) | ResNet18 | LRW500, 1000 | N/A | 84.1 |
| | TM-sep2seq (Afouras et al., 2018) | ResNet | LRS2-BBC | 49.8 | N/A |
| | TM-CTC (Afouras et al., 2018) | ResNet | LRS2-BBC | 65 | N/A |
| | STF (Zhang et al., 2019) | ResNet18 | LRW, LRS2,3 | 51.7 | 83.7 |
| | WAS (Chung et al., 2017) | Conv. | | 70.4 | 76.2 |
| | Perfect Match (Chung et al., 2019) | TC-5 | LRW | 71.6 | N/A |
| SSL | MoCo (He et al., 2020) | ResNet18 | K-AV-15K | 71.5 | 61.2 |
| | CPC (Oord et al., 2018) | ResNet18 | K-AV-15K | 66.7 | 65.3 |
| | DPC (Oord et al., 2018) | ResNet18 | K-AV-15K | 65.1 | 67.5 |
| | InfoMax (Hjelm & Bachman, 2020) | ResNet18 | K-AV-15K | 53.2 | 70.7 |
| | AVSlowFast (Xiao et al., 2020a) | ResNet18 | K-AV-15K | 56.1 | 75.8 |
| Ours | | ResNet18 | LRS2, 3 | 46.7 | 86.9 |
| | | ResNet18 | K-AV-15K | 47.8 | 83.7 |
| | | ResNet18 | K-AV-240K | **45.1** | **89.2** |

Table 1: Lip reading comparison with SOTA. LRW (top-1 accuracy, the higher the better) LRS (WER, the lower the better). The supervised approaches (Superv.) directly train the models on the dataset listed in the table. The self-supervised approaches (SSL) and Ours pretrain the models on the datasets listed in the table, and then finetune the model on LRW and LRS2 with the same protocol. Underline refers the best results from of the supervised approaches. Blue color highlights the comparison of ours with the self-supervised approaches under the same setting.

**Lip Reading**. Visually recognizing a speaker's utterance is useful and challenging task. Lip movements for different letters can be visually similar to each other (e.g., b and p, d and t). This requires the learned visual representation to contain fine-grained spatio-temporal information, rather than global semantics. In evaluating our pretrained model on the lip reading task, we focus on investigating whether our approach successfully learns fine-grained spatio-temporal visual information. For a fair comparison with state-of-the-art (SOTA) approaches, we use the same data processing protocol as Zhang et al. (2019). For LRW and LRS2, we detect 68 facial landmarks in each video frame using dlib (Castelli & Pagano, 2002). We use the outer eye and nose tip landmarks to align the detected face in each frame using an affine transform. Finally, an image of size $112 \times 112$ is cropped from the aligned face with the lip landmarks at the center. The cropping is such that the lips occupy $\frac{1}{3}$ of the image width. During finetuning, we apply random horizontal flipping. We concatenate the global

| Training | Model | Visual | Audio | Backbone | AUC |
|---|---|:---:|:---:|:---:|:---:|
| Superv. | Capsule (Nguyen et al., 2019b) | ✓ | | VGG-19 | 53.3 |
| | Multi-task (Nguyen et al., 2019a) | ✓ | | Y-shape | 53.6 |
| | HeadPose (Yang et al., 2019) | ✓ | | N/A | 55.9 |
| | Two-stream (Zhou et al., 2017) | ✓ | | Inception3 | 61.4 |
| | VA-MLP (Matern et al., 2019) | ✓ | | N/A | 61.9 |
| | VA-LogReg (Matern et al., 2019) | ✓ | | N/A | 66.2 |
| | Meso4 (Afchar et al., 2018) | ✓ | | Inception4 | 75.3 |
| | Xception-c40 (Rossler et al., 2019) | ✓ | | XCeption | 69.7 |
| | Xception-c23 (Rossler et al., 2019) | ✓ | | XCeption | 72.2 |
| | FWA (Li & Lyu, 2018) | ✓ | | N/A | 72.7 |
| | DSP-FWA (Li & Lyu, 2018) | ✓ | | N/A | 75.5 |
| | Siamese (Mittal et al.) | ✓ | ✓ | N/A | 84.4 |
| | MDS (Chugh et al., 2020) | ✓ | ✓ | ResNet18 | 91.5 |
| SSL | MoCo (He et al., 2020) | ✓ | | ResNet18 | 60.2 |
| | CPC (Oord et al., 2018) | ✓ | | ResNet18 | 67.9 |
| | DPC Han et al. (2019b) | ✓ | | ResNet18 | 71.2 |
| | InfoMax (Hjelm & Bachman, 2020) | ✓ | | ResNet18 | 85.3 |
| | AVSlowFast (Xiao et al., 2020a) | ✓ | | ResNet18 | 80.9 |
| Ours | (pretrain DFDC) | ✓ | | ResNet18 | 95.5 |
| | Multimodal (DFDC) | ✓ | ✓ | ResNet18 | 95.6 |
| | Ours (K-AV-15K) | ✓ | | ResNet18 | 90.1 |
| | Ours (K-AV-240K) | ✓ | | ResNet18 | **96.7** |

Table 2: Comparison with SOTA approaches on deepfake detection. For training on DFDC, we use the same training and testing list as Chugh et al. (2020). Mittal et al. use the same number of randomly sampled training data (180K). All the numbers of the other SOTA approaches are collected from Chugh et al. (2020). All the self-supervised approaches (SSL) are pretrained on K-AV-15K, and finetuned and tested on the same training and testing set of DFDC.

features and local features produced by our pretrained $E_v^g$ and $E_v^l$, respectively. Both encoders are finetuned with the whole model.

We compare our approach with SOTA supervised lip reading methods. For LRW, we evaluate on a 500-way word classification task and report top-1 accuracy. For LRS, we report the word error rate (WER). Table 1 shows that our approach outperforms SOTA approaches on both LRS and LRW by a large margin, i.e. 4.7% WER reduction on LRS and 5.1% accuracy improvements on LRW. These results show that our proposed approach can capture the fine-grained spatio-temporal information necessary for lip reading. We also compare our model with other self-supervised approaches with the same backbone and using the same pretraining dataset. Our approach outperforms these by a large margin and demonstrates the effectiveness of our proposed approach.

**Deepfake Detection**. Our hypothesis here is that deepfakes tend to be characterized by audio-visual inconsistencies such as a misalignment between lip motions and audio, unnatural facial and lip appearance/movements or asymmetry between facial regions such as the left and right eyes. Such artifacts could be detected through local spatio-temporal features. We use our pretrained model and finetune it on the DFDC dataset and evaluate performance using video-wise Area Under the Curve (AUC). We follow the same data preprocessing protocol as in SOTA approaches for this task. We perform face detection to crop the face region in each video frame. We concatenate the global and local visual features that are produced by our pretrained global and local visual encoders, and finetune the pretrained encoders with the whole model. The results are shown in Table 2. For a fair comparison, we use the same training and test video sets as Chugh et al. (2020). Among all the compared approaches, Chugh et al. (2020) and Mittal et al. use both visual and audio sequences, while the other approaches use only the visual sequences for detection. As we can see, when using only visual signal, our approach outperforms all previous SOTA approaches (AUC=96.7). We also compare our model with the other SOTA self-supervised approaches (shown in blue color). Again, our model outperform the best benchmark by a large margin (90.1 vs. 85.3).

**Action and Sound Classification.** To evaluate performance in learning discriminative global spatio-temporal representations, we use our pretrained model for action and sound classification. For action classification, we finetune both pretrained global and local visual encoders by concatenating the global and local representations. For audio classification, we finetune our pretrained audio encoder

$E_a$ with the audio classification model. To evaluate on action and audio classification, we compare both our models that were pretrained on Kinetics-700 and K-AV-240K with SOTA approaches pretrained on a dataset of the same scale (Kinetics). We find that on all three benchmarks, i.e. UCF101, HMDB51 and ESC50, our approach achieves a new SOTA (91.2% on UCF101, 61.9% on HMDB51 and 80.1% on ESC50) - see Table 3.

| | Method | Pretrain Dataset | UCF101 | HMDB51 | ESC50 |
|---|---|---|---|---|---|
| Sperv. | Scratch | - | 46.5 | 17.1 | N/A |
| | Scratch Random Forest (Piczak, 2015b) | - | N/A | N/A | 44.3 |
| | Scratch ConvNet (Piczak, 2015a) | - | N/A | N/A | 64.5 |
| | Scratch ConvRBM (Sailor et al., 2017) | - | N/A | N/A | 86.5 |
| | Supervised | ImageNet | 82.8 | 46.7 | N/A |
| SSL | RotNet3D (Jing & Tian, 2018) | K400 240K | 62.9 | 33.7 | N/A |
| | AVTS (Korbar et al., 2018) | K400 240K | 85.8 | 56.9 | 76.7 |
| | MotionPred (Wang et al., 2019) | K400 240K | 61.2 | 33.4 | N/A |
| | ST-Puzzle (Kim et al., 2019) | K400 240K | 65.8 | 33.7 | N/A |
| | ClipOrder (Xu et al., 2019) | K400 240K | 72.4 | 30.9 | N/A |
| | CBT (Sun et al., 2019) | K400 240K | 79.5 | 44.6 | N/A |
| | DPC (Han et al., 2019a) | K400 240K | 75.7 | 35.7 | N/A |
| | XDC (Alwassel et al., 2019) | K400 240K | 84.2 | 47.1 | 78.0 |
| | SeLaVi (Asano et al., 2020) | K400 240K | 83.1 | 47.1 | N/A |
| | AVID (Morgado et al., 2020) | K400 240K | 87.5 | 60.8 | 79.1 |
| | GDT (Patrick et al., 2020) | K400 | 89.3 | 60.0 | N/A |
| Ours | | K700 240K | **91.2** | **61.9** | 79.7 |
| | | K-AV 240K | 90.1 | 61.3 | **80.1** |

Table 3: Comparison of SOTA approaches on action classification and sound classification. We specify pretraining dataset and the number of samples used if they are reported in the original papers (N/A: not available). We highlight the **best** results and the second best results.

## 4.2 ABLATION AND ANALYSIS

**The importance of global-local contrast for local information needed tasks.** Here, we want to investigate specifically how the pretraining pretext task impacts local information needed for downstream tasks (i.e., lip reading and deepfake detection) and compare our task with those used in other work. We pretrain our model and the other state-of-the-art self-supervised pretraining approaches with the same backbone (3DResNet-18) and pretrain dataset (K-AV-15K). After pretraining, we finetune each model on the downstream benchmarks follow the same protocol. The results are listed in Table 4. As we can see that, when we only vary the pretext task during pretraining, our models outperform all the other SSL-based approaches by a large margin, which demonstrate the effectiveness of our proposed approach. We also find that InfoMax and AVSlowFast performs better than the others (MoCo, DPC and CPC). We believe this is because InfoMax draws more attention to the spatial local information and AVSlowFast is capable of learning more fine-grained temporal information, which are critical for the tasks of lip reading and deepfake detection. MoCo, which is successful for visual classification tasks fails in both lip reading and deepfake detection. This supports our argument that naïvely using an SSL approach may not achieve good performance for a large variety of different tasks.

**The roles of global and local information**. To demonstrate the importance of jointly learning global-local representations during pretraining, we evaluate a baseline model that was pretrained without the local contrastive objective (Ours w/o local cont.). As we can see from Table 5, when compared with the model which was pretrained using our full objective (Ours), the performance significantly drops on all the benchmarks. Optimizing only for global representations during pretraining generalizes poorly to the tasks that require local information. Note that, for a fair comparison, we only use the global features (Global Feat.) for each downstream tasks. Furthermore, we test whether using the local, global, or local and global features after pretraining yields better performance. We can see that, the best performance is achieved by utilizing both the global and local features and this is true for all the benchmarks. For comparison, we report the results achieved by the other SOTA approaches when using both the global and local features.

| Model | Pretext task | LRS | LRW | DFDC |
|---|---|---|---|---|
| MoCo (He et al., 2020) | Moment. Cont. | 71.5 | 61.2 | 60.2 |
| CPC (Oord et al., 2018) | Pred. Cont. | 66.7 | 65.3 | 67.9 |
| DPC (Han et al., 2019b) | Dense Pred. Cont. | 65.1 | 67.5 | 71.2 |
| InfoMax (Hjelm & Bachman, 2020) | Glo-loc NCE | 53.2 | 70.7 | 85.3 |
| AVTS (Korbar et al., 2018) | AVS | 72.1 | 64.9 | 63.1 |
| AVSlowFast (Xiao et al., 2020a) | AVS + Rot | 56.1 | 75.8 | 80.9 |
| Ours | Glo-loc MIL | **47.8** | **83.7** | **90.1** |

Table 4: Comparison with existing self-supervised pretraining approaches on lip reading and deep-fake detection. All the results are computed based on the implementation by us. We pretrain each SOTA model with the same pretext task proposed in their paper. "Moment. Cont.": momentum contrast; "Pred. Cont.": predictive contrast; "Dense. Pred. Cont.": dense predictive contrast; "Glo-loc NCE": global-local NCE; "AVS": audio-visual sync; "Rot": rotation detection

| Model | Feature | LRS | LRW | DFDC | UCF101 | HMDB51 |
|---|---|---|---|---|---|---|
| w/o loc. | Glob. | 70.9 | 65.3 | 67.9 | 82.3 | 57.1 |
| Ours | Glob. | 47.6 (↓ 23.3) | 86.8 (↑ 21.5) | 92.6 (↑ 24.7) | 89.2 (↑ 6.9) | 59.9 (↑ 2.8) |
| Ours | Loc. | 46.5 | 88.9 | 95.9 | 88.5 | 58.3 |
| Ours | Glob.+Loc. | **45.1** | **89.2** | **96.7** | **90.1** | **61.3** |

Table 5: The roles of global and local information on different benchmarks. "Ours": our model that pretrained with the whole objective. "Ours w/o loc.": our model that pretrained without the local contrastive objective. "Glob.": features extracted from the global encoder. "Loc.": features extracted from the local encoder. Underline refers to the best results from the other approaches.

| Local Contrast | LRS | LRW | DFDC | UCF101 | HMDB51 |
|---|---|---|---|---|---|
| w/o MIL | 40.4 | 79.2 | 88.9 | 87.8 | 56.3 |
| Ours | 47.8(↑ 7.4) | 83.7(↑ 4.5) | 90.1(↑ 1.2) | 88.1(↑ 0.3) | 56.8(↑ 0.5) |

Table 6: Comparison of models pretrained with and without using MIL for local contrast.

**The role of MIL-NCE in local contrastive objective**. When performing the local contrastive objective, we adopt the MIL-NCE as our loss function. Miech et al. (2020) also employed the MIL-NCE as the loss function to mitigate the misalignment in narrated videos. Different from their motivation, our goal here is to encourage fine-grained temporal alignment of the audio with video features. To validate its effectiveness, we evaluate an alternative without MIL-NCE as the local contrastive loss function. Specifically, we adopt an average temporal pooling on each window of the audio features and use the vanilla contrastive loss over the synchronized audio and visual features. The results are shown in Table 6. As we can see, when we perform the local contrast without the MIL-NCE objective, the performance on lip reading and deepfake detection drops considerably. While for activity classification, both loss function achieves comparable results.

**The role of attention**. Our pipeline allows us to leverage the global contrastive objective to capture local spatial information and use it as an attention map to assist local representation learning. Intuitively, our attention maps measure the amount of audio-visual correlation; such attention maps can highlight discriminative face regions useful for lip reading. We demonstrate the quality of our attention maps by replacing lip bounding boxes typically used in lip reading with our attention map. Specifically, instead of extracting features from the cropped lip/face region, we extract features from the entire frame (no lip/face cropping) and use our attention map to pool the features spatially. Note that the purpose of this experiment is to evaluate the quality of attention maps; we use audio signal just to obtain attention maps and discard it for word classification/deepfake detection. The results ("Ours Attention") show that it achieves results comparable to our best setting. On LRW and DFDC, it even outperforms SOTA approaches without relying on lip/face region detectors. It indicates that using global and local information in a collaborative way can yield good performance. In the lip reading task, local spatial information makes the local-contrast pathway pay more attention to lip movement, and thus achieves comparable effects of using a lip region detector. We also evaluate our model which finetuned directly on full frames. As we can see, when discarding the localized region

| Benchmark | SOTA | Our Best | Ours Full frame | Ours Attention |
|---|---|---|---|---|
| LRS | 49.8 TM-seq2seq | 45.1 | 71.9 | 51.2 No lip crop |
| LRW | 84.1 DFTN | 89.2 | 62.3 | 85.1 No lip crop |
| DFDC | 85.3 InfoMax V | 96.7 V | 68.1 V | 95.9 No face crop |
| | 91.5 MDS V+A | 97.1 V+A | – | |

Table 7: Evaluation on the role of attention mechanism in our approach. "V" uses only visual sequence and "V+A" uses both visual and audio sequence for finetuning on downstream tasks.

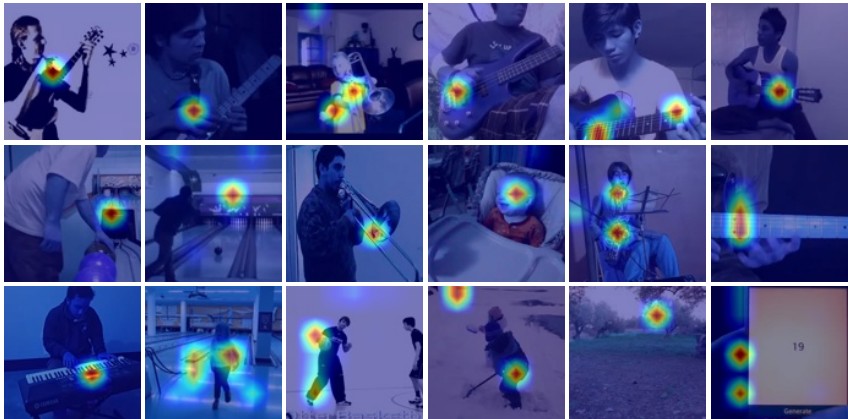

Figure 3: Visualization attention maps showing the audio-visual correlations in learned representations.

achieved either by detectors or attention mechanism, the performance significantly drops on all three benchmarks. It further validates the critical role of the attention mechanism in our approach.

**Interpretation of the learned representation.** To investigate how well local spatial information is captured through the global audio-visual contrast, we visualize the attention maps that induced by the pretrained audio and global video encoders. Such visualization can also be considered as performing sound source localization, i.e. locate objects that making sound. To achieve this goal, the network should capture the audio-visual correlation in a spatio-temporal grid. We thus use the attention map obtained by our pretrained model to visualize the sounding source in each frame. To investigate further, we use the Kinetics-sound (Carreira et al., 2019) dataset as videos in the dataset generally have a high-level of audio-visual correspondence. Specifically, we add another softmax layer on the obtained attention map, and then do bilinear interpolation of the $16 \times 16$ attention map back to the original image size, i.e. $192 \times 192$.

Fig. 3 shows that our learned attention maps tend to localize sounding sources in videos accurately. Especially, when visual content is highly related to the corresponding audio signal, our model performs particularly well. For example, the first row (frames from "playing instruments" videos) shows that our model can precisely localize the sounding region. For the other activities, like "baby talking," "playing basketball," "running," our model highlights regions with humans. We find that the attention map incorrectly highlights regions on samples that have ambiguous audio-visual relation. We show failure cases in the last two frames of the third row. As we can see, there is no visual content that clearly relates with the audio signal, and thus the model fails to find sounding sources.

## 5    CONCLUSION

We have presented a contrastive self-supervised approach to learning global and local audio-visual representations. Using audio and low-sampled and high-sampled video sequences as separate "views" of the data, we find that the learned representations can generalize well to tasks that involve global semantic understanding and fine-grained spatio-temporal understanding. We perform experiments on lip reading, deepfake detection, sound source localization, and action/sound classification tasks and in each case achieve strong results.

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
