# OpenReview forum: "Contrastive Self-Supervised Learning of Global-Local Audio-Visual Representations"
_ICLR.cc/2021/Conference — Reject_

### Official Review · AnonReviewer2 · 2020-10-26
**Well motivated approach, great results**

**Rating:** 7
**Confidence:** 4

**Review:**

**Summary**
This paper presents a new method of using self-supervised contrastive learning to learn global-local audio-visual representations, resulting in strong results on a variety of tasks. The proposed method leverages simultaneously captured information in audio and visual modalities at different time scales to generate different "views" of the data for contrastive learning.

**Strengths**
+ Learning general audio-visual representations by jointly optimizing both global-scale and local-scale objectives is well-motivated, and to the best of my knowledge has not been done before. The presented method of generating matching and contrasting "views" is conceptually simple, yet seemingly very effective.
+ Strong empirical results on a variety of downstream tasks indicate that the learned representations do indeed generalize well to both global and local downstream tasks.
+ The paper reads well, and method is easy to understand.

**Weaknesses**
- A better explanation about the motivation behind using the non-standard MIL-NCE training objective would be helpful.
- I'm not convinced of the need for MIL-NCE for the local contrastive objective. The original motivation behind the use of this objective was misalignment in narrated videos, which is a notorious problem in visual-textual mapping. Here, I'm not sure that there exists a "strict temporal mapping problem", as the the modalities are captured simultaneously, and should therefore be temporally synchronized. Perhaps performing a fixed/learned pooling of multiple audio features which correspond to the time span of a single visual feature, and optimizing using a more conventional contrastive loss would give better results.
- The role of spatial attention is evaluated in Sec. 4.1 and 4.2 by using the attention to extract an ROI, instead of standard lip/face cropping. Adding the results of using the full face frame as a baseline to the comparisons would give a more complete picture of the power of the spatial attention.
- The experiment section is lacking explanations of how to use the network/representations described in Sec. 3 to solve the downstream tasks. For example, how do you get from learned representations to LRW word classification, LRS WER, etc.?

Another related work which is worth referencing in Sec. 2 is "Audiovisual SlowFast Networks for Video Recognition" by Xiao et al, which is based on the original SlowFast visual-only work, and which also learns from a given video at two different sample rates, along with audio.

This is a good paper, in which a novel method for solving an important problem is proposed, and the resulting performance on tasks of interest is strong. I would like to see it accepted to ICLR, provided the above concerns are addressed.

---

> ### Author Response · Authors · 2020-11-23
> **Thanks for the positive comments! We clarified the need for MIL-NCE and added extra results to the revision.**
>
> We appreciate your thoughtful and constructive feedback (and a recommendation to accept our paper!).
>
> * **MIL-NCE in the local contrastive objective:** We agree that it may be unnecessary to deal with temporal mapping at the raw signal level since audio-visual streams are captured simultaneously. Our notion of temporal mapping is actually at the semantic embedding level, where the two streams may not be strictly aligned (e.g., a person picks up the phone only after the bell rings, not simultaneously). Our intuition is that giving the model the leeway to match audio-visual streams within a temporal window would allow it to learn more semantically discriminative features. To show this empirically, we conducted an ablation by pretraining our model with and without MIL-NCE for local contrastive loss. We find that the model pretrained with MIL-NCE consistently achieves a better performance across all five benchmarks (see Table 5). We have added the new results to the revision; see “The role of MIL-NCE in local contrastive objective” in Section 4.
>
> * **Full face baseline:** Thanks for this great suggestion! We have included this baseline in Table 7 (Ours full frame), which further demonstrates the power of our spatial attention approach.
>
> * We have added detailed explanations of how we use the pretrained model for each downstream task; see Section 4.1.
>
> * We have added AVSlowFast in Section 2 and included AVSlowFast baseline results in Tables 1, 2 and 4.

---

### Official Review · AnonReviewer3 · 2020-10-27
**fair idea with strong experiments**

**Rating:** 5
**Confidence:** 4

**Review:**

Summary: This paper proposes an audio-visual self-supervised learning approach based on two cross-modal contrastive loss that learns audio-visual representations that can generalize to both the tasks which require global semantic information and localized spatio-temporal information. Extensive experiments on 4 task demonstrate the usefulness of the learned representation.

Strengths:
+ The paper is nicely written and well motivated. Existing works tend to learn either global representations or local representations, while this work aims at learning versatile representations that generalize well to both scenarios.
+ Good observation and analysis to construct triplets from different spatial and temporal span to capture either global information or local information. A tailored global-local contrastive learning network is desgined to realize the proposed idea.
+ Extensive experiments are performed on 4 task across a number of datasets to demonstrate the generality of the learned representations.

Weakness:
- Among the tasks evaluated on, only Table 3 compares to prior self-supervised learning approaches. This demontrates the proposed method learns better global representations, but no results are shown to prove that the prior ssl methods fail to learn local representations. In Table 1 and Table 2, the method only compares to prior state-of-the-art methods of the corresponding task. It would be more convincing to compare to the representations of some of the methods in Table 3.
- For sound source localization, only some qualitative results are shown. Why not showing the quantitative results (IOU)? Only a figure of selected qualitative results is not convincing to this reviewer.
- For Table 1 and Table 2,  the prior methods all use different network settings from the proposed network. How to tell whether the  gain is from a better network architecture or the proposed global-local representation learning method? Without comparing in an apples-to-apples manner with other self-supervised contrastive learning methods (e.g., GDT, AVID, etc.),  it would be unconvincing that the proposed global-local audio-visual contrastive learning method indeed learns better representations that have the suggested properties.
- The related work is somewhat short and unorganized. It would be much better to re-organize the related work section (contrastive self-supervised representation learning, audio-visual learing,  etc. ) and highlight the differences to each group.

Justification of rating:
The paper proposes a decent idea to learin global-local reprentations and evaluate on various tasks/datasets. However, some necessary comparisons are missing in order to demonstrate the effectivenss of the proposed method. This reviewer is happy to raise the score if the rebuttal can clarify the concerns.

Post-rebuttal:
Thanks for the clarifications in the rebuttal. It addresses some of my concerns. However, I am still concerned about the unfair comparisons of baselines using different settings, and the newly added ssl baselines all outform the proposed method on LRS by a large margin. Therefore, I would like to keep my original rating.

---

> ### Author Response · Authors · 2020-11-23
> **We have added additional results to address your concerns.**
>
> * **Prior SSL methods on local representations:** We appreciate this great suggestion! We have added new results (see Table 4) comparing ours to six existing SSL pretext tasks on lip reading and deepfake detection that rely on local representations. In this experiment, we used the same backbone and the same datasets so that the comparison is apples-to-apples.
>
> * **Sound source localization:** We agree that showing quantitative results (e.g., IoU) would be much more convincing than qualitative results. To clarify, at the time of submission, our intention was to provide insights into why our method outperforms the baselines on the “local” tasks through some visualization, rather than tackling sound source localization as a task itself. We have now removed the section “Sound Source Localization” and moved the figure under Sec. 4.2. Ablation and Analysis, clearly mentioning that the figure isn’t meant to provide a full picture of how our method will perform on the task in general. Moving forward, we believe that the reviewer made an excellent point here and we promise to evaluate our method on sound source localization using available datasets with bounding box annotations of sounding source (e.g., https://github.com/DTaoo/Discriminative-Sounding-Objects-Localization).
>
> * **Different backbones:** The new Table 4 we have added to the revision reports results under the exact same setting, including backbone, dataset, data augmentation pipeline, optimizer, learning rate schedule, etc.; the only difference is the pretext task. We hope this will help address the reviewer’s concern. In general, this is unfortunately a common issue to the video self-supervised learning literature (i.e., if one method outperforms another, it is not clear whether it comes from a different backbone, a different pretext task, a different optimizer, or ...). We hope the community will find a solution to standardize the evaluation protocol.
>
> * **Related Work:** Thanks for this suggestion. We have reorganized the section and added a discussion about how our approach differs from existing work.

---

### Official Review · AnonReviewer1 · 2020-10-27
**Nice idea but very careful selection of pre-training dataset for each downstream task.**

**Rating:** 6
**Confidence:** 4

**Review:**

This paper proposes a self-supervised approach for audio-visual representation learning which consists of two losses, a global contrastive loss and a local contrastive loss. Each loss operates on differently sampled visual features.

Strengths:

- The method and experiments are described clearly.
- The idea to combine both local and global information in a self-supervised objective while pre-training a model is a sensible one.
- The performance on some downstream evaluations - eg.  lip reading as well as action classification on HMDB exceed SOTA comfortably.

Weaknesses
- The main motivation for the paper appears to be a self-supervised approach that can capture both global and local information, and hence the “views” of the data do not need to be carefully selected for every downstream task. However, it is worthwhile noting that the pre-training dataset is carefully chosen for each task: LRS for lip reading, DFDC for deep-fake detection and Kinetics for classification tasks. These pre-training datasets themselves capture important invariances in the way the datasets have been constructed.
- Training both contrastive losses jointly doesn’t seem to help performance -  the local contrastive loss is used for lip reading (Table 1: adding the global loss only shows 0.5% improvement for LRS and actually harms for LRW) while the global loss is sufficient for classification (adding in the local loss shows 0.2% in Table 3). Since the authors have already carefully selected the pre-training dataset for each downstream task, isn’t it also possible to just select which loss is needed based on the downstream task? I believe it would be really great if it were possible to train a single model on a large dataset without knowing the downstream task, and apply the same model to both a downstream task that requires “local information” and another downstream task that requires “global information”

Questions:
- Table 2 says 95.6% for multimodal but the text says performance drops to 94.6%, is this a typo? What is the performance on the DeepFakes downstream task with only the local or the global contrastive loss?
- For lip reading, why not pre-train on larger, “in the wild” datasets such as VoxCeleb2? LRS has been carefully curated as a lip reading dataset already. Then the same model could be used for a number of tasks such as deepfake detection, lip reading, speaker verification etc.


Suggestion for future work: (not taken into account for the review)
- Sec 4.3, is there a way to quantitatively assess sound source localisation performance, eg using datasets such as SoundNet-Flickr or AudioSet-instrument, as shown here: https://arxiv.org/abs/2007.06355
- It would be good to evaluate the audio representations on VGG-Sound (http://www.robots.ox.ac.uk/~vgg/data/vggsound/)  as well. Another interesting direction would be to compare to this work in the speech domain (https://ieeexplore.ieee.org/stamp/stamp.jsp?tp=&arnumber=9054057), since it also trains with two self-supervised contrastive losses, one trying to capture global representations for identity, and one trying to capture local representations (in time) for content. The downstream task here would be speaker verification.
- Table 1, some works are missing such as https://arxiv.org/pdf/1809.08001.pdf

---

> ### Author Response · Authors · 2020-11-23
> **New results of a single pretrained model show improved performances on downstream tasks.**
>
> We appreciate the positive feedback and detailed suggestions to improve our paper!  We have carefully incorporated your comments into our revision. The main updates are summarized below:
>
> * **The “one pretrained model to rule them all” idea:** Thanks for suggesting this great idea! In the submitted version, we did the domain-specific pretraining to allow for fair comparisons to prior work, but we cannot agree more that this new experiment will further demonstrate the generality of our approach. We have conducted this experiment and included the results in the revision. Here’s a summary of the experiment:
>
>     * We construct a new pretraining dataset by taking 120K videos from Kinetics-700 and 120K videos from AVSpeech, for a total of 240K videos (random sampled).
>
>     * We pretrain our model on this dataset and finetune it on different downstream tasks.
>
>     * This outperforms our previous results and achieves new SOTA results in lip reading, deepfake detection and audio event classification (see Tables 1, 2, 3 and 4). In activity classification, this new model achieves comparable results with the one pretrained on Kinetics-700. These results suggest that our model is robust on variations of the pretraining data and can effectively learn good representations.
>
>     * We also added new ablation results to Table 5, which shows the individual contributions of our local and global contrastive losses.
>
> * **95.6% (Table 2) vs. 94.6% (text):** Thanks for spotting this; it should be 95.6%.
>
> * **Local vs. Global:** The performance on the DeepFakes was 76.1% with only the local loss and 67.9% with only the global loss. We have included the global-only results across five downstream datasets in Table 5.
>
> * **VoxCeleb2:** This is also a great suggestion. Considering AVSpeech is a large-scale in-the-wild dataset, we believe that our new results with Kinetics-700 + AVSpeech would be sufficient to show this.
>
> * **Evaluating sound source localization** (on SoundNet-Flickr or AudioSet-instrument): We love this suggestion! We unfortunately couldn’t obtain those datasets in time for the rebuttal period, but we promise to add this to our camera-ready.
>
> * We have added the Perfect Match (https://arxiv.org/pdf/1809.08001.pdf) results to Table 1.

---

### Official Review · AnonReviewer4 · 2020-10-30
**Interesting direction, but lacking in evaluation and limited novelty**

**Rating:** 5
**Confidence:** 4

**Review:**

Summary: This paper presents a new contrastive audio-visual learning method. Like previous work, they use self-supervision to learn a video feature set by training a network to associate audio and visual "views" taken from the same video. Their main contribution is to jointly learn from both "local" and "global" information. They simultaneously optimize two contrastive objectives. First, there is a global objective, which computes a feature set using a low framerate video, pools over time, and obtains negatives from other different videos. Second, there is a local contrastive objective that uses a higher framerate video, pools over space but not time, and gets negatives from other timesteps of the video. They optimize both losses jointly using a spatially-aware pooling method that provides information from the global pathway to the local pathway. They compute attention by taking dot products between the visual and audio features, and using this attention to pool local visual features (instead of a global pooling).

Pros:
- The paper evaluates the model on a diverse set of downstream tasks: lip reading, deepfake detection, action recognition, and sound classification.
- The model performs well on the downstream tasks (competitive or better than other self-supervised pretraining methods).
- The idea of designing new network architectures that can take advantage of local and global information is an interesting direction.

Cons:
- There is not very much novelty in the loss functions. The idea of using losses that sample negative examples from two sources (other videos and other timesteps) has been used in other work such as (Korbar et al. 2018). The main novelty, instead, is the network architecture.
- The evaluation for the architecture is quite minimal. There are a few ablations in each task, but they aren't chosen very systematically. These ablations also seem to simultaneously change the architecture and the loss, which makes it hard to take much away from them.
- The paper relies exclusively on pretraining + finetuning experiments. While it is encouraging that the method outperforms other self-supervised pretraining methods, these comparisons are not completely apples-to-apples, since they use different network backbones. This is to some extent unavoidable, since this is the standard practice in the video self-supervision area. However, since this is one of the only forms of evaluation (and there are few ablations) it makes it hard to gauge the effectiveness of the method.
- The accuracy improvements from using the local contrastive loss are confined to the lipreading task. It does not seem to improve the model on the action recognition task.
- The evaluation of the localization task is quite minimal. They show a few qualitative results without comparing to other methods. The dot-product attention used in the model is quite similar to previous methods, such as (Arandjelović and Zisserman, 2018).  The paper should explain what the novelty of their approach is, and should compare with previous methods.
- I didn't find the paper to be particularly well written. I found it challenging to understand the motivation for the method and the description of it.

Overall:
While this is a promising direction, the evaluation of the architecture (the main novelty) is not very thorough.

---

> ### Author Response · Authors · 2020-11-23
> **We have clarified our novelty and added extra experimental results**
>
> We appreciate the reviewer for such thoughtful and detailed comments.
>
> * **Novelty:** We understand the reviewer’s disappointment with our loss functions. We believe that our main novelty is in the idea of combining global and local objectives to learn representations that generalize to both the tasks that require global semantic information (classification) and the tasks that require fine-grained spatio-temporal information (localization). As ours is the first to propose this idea, we used well-established loss functions to clearly demonstrate that any improvements come from our joint global-local formulation, not from special algorithmic treatments to each of the loss formulations. To the best of our knowledge, our work is the first to demonstrate that a single pretrained model can generalize to both classification and localization tasks (please also see our response to R1, “one pretrained model to rule them all”), which substantiates our novelty.
>
> * **Different backbones:** Please correct us if we misinterpreted the reviewer’s concern -- Reading between the lines (from the second and the third bullet points under “Cons”), we believe the reviewer is referring to various downstream evaluations and ablations based on different backbone architectures. This is a valid point. We agree that this makes it difficult to gauge the effectiveness of our method. To address this, we have added new experimental results to the revision: 1) pretrain a single model and then transfer it to different downstream tasks, thus using the same backbone architecture across different tasks; see our response to R1 “one pretrained model to rule them all”; 2) compare the effectiveness of different pretext tasks while keeping all the other variables the same (backbone network, data, optimizer, etc.); see our response to R3 “different backbones.”
>
> * **Local contrastive loss for action recognition:** We want to clarify that our model achieves new SOTA results in action recognition (UCF101 and HMDB51) and sound classification (ESC50) by using our proposed global-local joint training objective, not by using the global loss alone. To show the importance of the joint training objective, we have conducted a new ablation where we pretrain models with and without the local contrastive loss; see Table 5 in the revision). The new results show that performance drops significantly across all five downstream datasets. We have added an extended discussion of the two contrastive objectives in Section 4.2 “The roles of global and local information.”
>
> * **Sound source localization:** As we noted in our response to R3 “Sound source localization,” we believe that this is a great suggestion. We agree with R3 that a quantitative evaluation (IoU) would be much more convincing, and we promise to include this result in our final version.
>
> * **Writing:** We did a major overhaul of the paper and carefully checked to make sure that the logic flows smoothly.

---

### Decision · Program_Chairs · 2021-01-07
**Final Decision**

**Decision:**

Reject

**Comment:**

The paper received mixed reviews. While AnonReviewer1 and AnonReviewer2 liked the idea of jointly learning global-local representations, the other reviewers were concerned about the technical novelty. Reviewers also raised various questions about the experiments and ablation studies. AC found that the rebuttal well addressed the reviewers' questions about the experiments, but it failed to elaborate on the "why" of combining global and local self-supervised representations. AC agreed with AnonReviewer3 and AnonReviewer4's concerns on technical novelty. Considering the reviews, we regret that the paper cannot be recommended for acceptance at this time.  The authors are encouraged to consider the reviewers' comments when revising the paper for submission elsewhere.